# Biochemical and Biotechnological Insights into Fungus-Plant Interactions for Enhanced Sustainable Agricultural and Industrial Processes

**DOI:** 10.3390/plants12142688

**Published:** 2023-07-19

**Authors:** Anderson Giehl, Angela Alves dos Santos, Rafael Dorighello Cadamuro, Viviani Tadioto, Iara Zanella Guterres, Isabella Dai Prá Zuchi, Gabriel do Amaral Minussi, Gislaine Fongaro, Izabella Thais Silva, Sergio Luiz Alves

**Affiliations:** 1Laboratory of Yeast Biochemistry, Federal University of Fronteira Sul, Chapecó 89815-899, SC, Brazil; 2Graduate Program in Biotechnology and Biosciences, Federal University of Santa Catarina, Florianópolis 88040-900, SC, Brazil; 3Laboratory of Applied Virology, Department of Microbiology, Immunology and Parasitology, Federal University of Santa Catarina, Florianópolis 88040-900, SC, Brazil; 4Graduate Program in Pharmacy, Federal University of Santa Catarina, Florianópolis 88040-900, SC, Brazil; 5Graduate Program in Environment and Sustainable Technologies, Federal University of Fronteira Sul, Cerro Largo 97900-000, RS, Brazil

**Keywords:** biocontrol, bioherbicide, mold, phytohormone, phytopathogen, plant-growth promoter, yeast

## Abstract

The literature is full of studies reporting environmental and health issues related to using traditional pesticides in food production and storage. Fortunately, alternatives have arisen in the last few decades, showing that organic agriculture is possible and economically feasible. And in this scenario, fungi may be helpful. In the natural environment, when associated with plants, these microorganisms offer plant-growth-promoting molecules, facilitate plant nutrient uptake, and antagonize phytopathogens. It is true that fungi can also be phytopathogenic, but even they can benefit agriculture in some way—since pathogenicity is species-specific, these fungi are shown to be useful against weeds (as bioherbicides). Finally, plant-associated yeasts and molds are natural biofactories, and the metabolites they produce while dwelling in leaves, flowers, roots, or the rhizosphere have the potential to be employed in different industrial activities. By addressing all these subjects, this manuscript comprehensively reviews the biotechnological uses of plant-associated fungi and, in addition, aims to sensitize academics, researchers, and investors to new alternatives for healthier and more environmentally friendly production processes.

## 1. Introduction

Molds and yeasts occupy different spaces in ecosystems, from the soil to the atmosphere, participating in important ecological relationships such as nutrient cycling [1]. In many environments, these fungi establish symbiosis with plants. For the last 400 million years (at least), their relationship has sometimes been harmful or beneficial to plants. In fact, since the existence of fungi predates that of the first plant species, fungi were fundamental for the colonization of the terrestrial environment by plants [2]. An important example of how this association contributed to the development of plant biomass is in the interaction between angiosperms and yeasts (unicellular fungi). In this mutualism, yeasts metabolize sugars and amino acids present in flower nectar and produce volatile organic compounds (VOCs) that attract pollinators. Animals, in this way attracted, provide plant reproduction [3].

During this terrestrial-environment colonization process, plants adapted both morphologically and genetically, starting to present receptors capable of identifying molecular patterns of mutualistic fungi and pathogens, such as Microbe- and Pathogen-Associated Molecular Patterns (MAMPs and PAMPs, respectively) [4]. Complementarily, filamentous fungi and yeasts began to adapt to the conditions of symbiosis with plants, being exposed to variations in temperature, solar radiation, humidity, and nutrient access [1].

Usually, when we think about the association of fungi with plants, we remember the damage caused to crops by invading fungi, which accounts for about 10 to 20% of crop losses and causes about 100 to 200 billion dollars a year in damage [5]. However, of the 150,000 fungi described by science, only 8000 are phytopathogenic [6,7]. Most of this fungus-plant relationship is beneficial, such as mycorrhizal fungi that provide, for example, nitrogen and phosphorus. In addition, fungi can stimulate plant growth and fix iron by producing siderophores [8,9].

In endophytis, the fungus lives in the host tissue, producing compounds that can promote plant tolerance to stress. Furthermore, endophytic fungi can inhibit pathogens and pests through their VOCs, enzymes, and bioactive compounds [10,11,12]. Interestingly, recent studies have shown the potential use of antagonistic fungi as a biocontrol of pests in agriculture, even inhibiting the growth of parasitic fungi (thanks to the mycotoxins produced by the endophytic part). In this case, fungi serve as an active agent when applying the live microorganism or its fermentative substrate [13,14,15]. This strategy enables the replacement of chemical agents that cause environmental damage and, thus, provides compliance with SDG 2 of the United Nations.

Moreover, plant-associated fungi have been investigated for different biotechnological purposes. Although some species are widely studied and applied in different industrial sectors, many new non-conventional molds and yeasts have shown biotechnological potential as well. In such cases, applications are vast for food, pharmaceutical, or fuel companies [3,10,16].

Through a biochemical and ecological look at fungus-plant interaction, this article presents the compounds involved in mutualistic relationships, whether in promoting plant growth, pest control, or pathogenic interaction with the plant. Finally, the biotechnological importance of prospecting fungi associated with plants for commercial applications is discussed here. As a clarifying note, herein we are considering molds or filamentous fungi as pluricellular microorganisms of the Kingdom Fungi with hyphal and mycelial growth and yeasts as unicellular fungi that grow as individual cells or form pseudohyphae. Another important definition: we considered volatile compounds to be those low-molecular-weight solids or liquids that easily reach the gas phase at atmospheric pressure and room temperature.

## 2. Mutualistic Interaction between Fungi and Plants

In many cases, mutualistic interaction with fungi is vital for plants. This is the case with mycorrhizal fungi, for instance. However, analyses of plant-associated molds and yeasts reveal many more benefits for both parts and for food production. Fungi may produce metabolites associated with promoting plant growth. These molecules can function as hormones, macro- and micronutrients, organic acids, siderophores, lytic enzymes, and/or volatile compounds, exerting direct or indirect action on plant growth [17,18,19,20]. Moreover, yeasts and molds can act as plant-protective agents, preventing plants from being infected with pathogens (Figure 1).

### 2.1. Plant Growth Promotion

#### 2.1.1. Yeasts

As part of yeast secretomes, siderophores have a metallic affinity for iron and zinc. The chelation of these micronutrients promotes their adsorption in the liquid fraction of the soil and, consequently, the growth of plants. The species *Pseudozyma aphidis* is recognized for its production of siderophores. Likewise, the species *Metschnikowia pulcherrima*, *Papiliotrema flavecens*, and *Rhodotorula glutinis* are efficient in assimilating iron in ferric form from the surrounding substrate, even under metal-limiting conditions. Similarly, *Candida pimensis*, *Candida apicola*, and *Dothideomycetes* sp. have already been shown to be able to increase zinc availability in plants [19,20,21]. 

In addition to the chelation of metal ions, yeasts can also contribute to the solubilization of phosphate, which is usually poorly available to plants despite being necessary for their growth. Millan et al. [20], when isolating yeasts from Spanish vineyards, verified that the bioavailability of phosphate was higher in the presence of the yeast *Lachancea thermotolerans*. The ability of yeasts to increase the bioavailability of nutrients is linked to the secretion of acids, enzymes, and chelators [22,23,24]. In a study with yeasts isolated from different tissues of red and blue creole maize, the best species for phosphate solubilization were *Rhodotorula mucilaginosa*, *Clavispora lusitaniae*, *Suhomyces prunicola*, and *Kurtzmaniella quercitrusa* [21].

Isolated from the leaf surface of *Drosera indica*, the yeasts *Hannaella coprosmaensis*, *Aureobasidium pullulans*, and *Pseudozyma aphidis* increased the number of lateral roots by inhibiting fungal pathogens and by producing the phytohormone indole-3-acetic acid (IAA) [25]. IAA activates agile and long-lasting responses in plants [26,27] and regulates several of their physiological processes, which is why it has been proposed as a possible biofertilizer [28]. Furthermore, IAA can act in signaling interactions between microorganisms and plants in a reciprocal way [8]. Also isolated from leaves (in this case, from *Drosera spatulata*), the yeast *Sporidiobolus ruineniae*, thanks to the production of IAA, increased primary root formation, stem length, chlorophyll content, and the number of lateral roots and leaves [19]. IAA was also found in the filtrate of the yeasts *Kazachstania rupicola* and *Rhodosporidium diabovatum*, which were isolated from the water of a bromeliad tank. When this filtrate was applied to bromeliads, a higher photosynthetic rate, and greater plant growth were observed [29]. Similarly, wild yeasts isolated from a Mexican variety of maize produced IAA and caused a 1.5-fold increase in height and root length in inoculated-maize plants compared to non-inoculated controls [21].

The yeasts *Pichia kudriavzevii* and *Issatchenkia terricola* isolated from grape, pomegranate, tomato, and black grass seeds were reported to produce, besides siderophores and IAA, hydrogen cyanide (a volatile organic compound associated with aromas from plants) and auxin [30]. Auxin is another essential plant hormone for developing stems, roots, and fruits. This phytohormone also acts on the movement of the plant in search of a better use of the sun [31]. The yeasts *Solicoccozyma* sp., *C. lusitaniae*, *R. glutinis*, and *Naganishia* sp. are also efficient producers of auxins. When inoculated into the plant *Arabidopsis thaliana*, they induced the expression of auxin-responsive genes. Inoculation of auxin-producing yeasts in maize plants improved height, fresh mass, and root length [21].

Millan et al. [20] demonstrated that yeasts of the genus *Meyerozyma* stand out as ammonia producers, which serves as a probable communication signal between colonies [32]. At the same time, nitrogen availability through secreted ammonia promotes the growth of plants that benefit from this nitrogenous macronutrient, which is essential for plant development [33]. 

The yeasts *Meyerozyma guilliermondii* and *Pichia dianae* isolated from grapes provided an increase in shoot and root weight, the formation of long root hairs, and better branching. These yeasts also increased the chlorophyll content of the leaves [20]. In the study by Ramos-Garza et al. [34], yeasts were isolated from the rhizosphere of plants cultivated in soil with heavy metals, and the isolates obtained from *R. mucilaginosa* and *Cystobasidium sloffiae* promoted plant growth and seed germination in a bioassay using brown mustard (*Brassica juncea*). 

By protecting plants from pathogens, yeasts can also contribute to plant growth. The fungus *Fusarium oxysporum*, which is responsible for plant diseases that lead to reduced growth, colonizes phytoenvironments and can settle inside the seeds. The yeasts *Candida orthopsilosis* and *R. mucilaginosa* managed to inhibit the growth of this fungus and thus reduce the mycotoxin zearalenone produced by it within wheat plants. In wheat roots and ears infected with *F. oxysporum*, inhibition of zearalenone was complete when treated with both yeasts [35]. 

Yeasts can also associate with other microorganisms to promote plant growth. Isolated from the bean roots, the yeast *Candida tropicalis* and the bacterium *Rhizobium* sp., when inoculated together, promoted greater nodulation and absorption of nutrients by the plants, which showed increased growth. *Candida tropicalis* enhances nodulation through its unique metabolites, such as indole compounds, tryptophan, phenolic compounds, and α-D-galactopyranoside [36].

#### 2.1.2. Molds

Fungi possess exceptional metabolic abilities, which have led to the development of several biotechnological compounds that have significantly impacted the fields of medicine and different industry sectors. These compounds encompass a wide range of products, such as vitamins, plasma substitutes, anticancer agents, healing accelerators, enzymes, proteins, polysaccharides, and organic acids [37,38,39]. Particularly, filamentous fungi such as *Trichoderma*, *Penicillium*, and *Aspergillus* have been extensively investigated for their ability to promote plant growth. These fungi can colonize and interact with plant roots in several ways, including nutrient absorption enhancement, phytohormone synthesis, and the induction of systemic resistance and abiotic stress tolerance. As biological control agents, they protect plants by competing for resources and space while producing antibiotics and hydrolytic enzymes that hinder the growth of phytopathogens. These fungi have found utility in various crops, including tomatoes, corn, beans, soybeans, and lettuce. An additional benefit of these fungi is their ubiquitous presence and widespread distribution in soil [40,41,42].

The relationship between *Trichoderma atroviride* and *Arabidopsis thaliana* was studied using Petri plates with co-cultivation of this plant and the endophytic fungus [43]. Due to volatile emissions, it was observed that *T. atroviridae* can induce lateral root proliferation and enhance biomass production. Besides, it was identified that sucrose biosynthesis is induced after *Trichoderma* cultivation. It is suggested that this induction occurs because of the volatiles produced by the fungus, which act through transporters like SUC2 and SWEET to trigger root growth [44]. Indeed, the growth promotion of some plants by *Trichoderma* spp. may be due to several factors, including their ability to acidify the rhizosphere, solubilize nutrients, and produce secondary metabolites such as blends of volatile substances [45]. Some VOCs emitted by *Trichoderma* include lactones, ketones, alcohols, mono- and sesquiterpenes, esters, and aldehydes—compounds produced through various metabolic pathways [46].

A study with strains of dark septate endophytes (DSEs), *Cadophora*, *Leptodontidium*, *Phialophora*, and *Phialocephala*, isolated from roots of poplar trees from metal-polluted sites, demonstrated that root and shoot growth promotion is plant- and strain-dependent, with *Phialophora* and *Leptodontidium* strains having a higher potential for improved plant growth. The release of VOCs was related to six strains, but the production of auxins was associated with all of them. *Cadophora* and *Leptodontidium* strains demonstrated the ability to grow and produce VOCs even at high concentrations of several metals, which suggests that endophytes present significant potential to promote sustainable production of bioenergy crops in the phytomanagement of sites with metals [47].

In a more general sense, given the broad host specificity exhibited by these fungi, they could potentially serve as plant growth-promoting agents in horticulture and agriculture. When analyzing the plant-growth-promoting activity of 15 strains of endophytic fungi isolated from *Sophora flavescens* (belonging to the genera *Alternaria*, *Didymella*, *Fusarium*, and *Xylogone*), Turbat et al. [48] verified that all the isolated strains produced IAA, five demonstrated phosphate solubilization activities, and twelve secreted siderophores. According to the literature, root elongation is influenced by exogenous IAA in a concentration-dependent manner [49]. Similarly, Ismail et al. [50] also found that strains of *Alternaria sorghi* and *Penicillium commune* were able to produce IAA. The authors compared the direct application of plant hormones to bean crops with the application of fungi and, interestingly, observed that cultures inoculated with fungi outperformed those treated with exogenous hormones in characteristics such as increased biomass and carbohydrate and protein content.

In addition to acting directly on the growth of some plant crops, fungi can also act positively through indirect mechanisms, such as increasing tolerance to abiotic stresses, mainly salinity and drought, and resistance to soils with heavy metals. For example, under conditions of high salinity, studies show that the endophytic fungus *Epichloe bromicola* contributes to the tolerance of wild barley (*Hordeum brevisubulatum*) to this stress, increasing germination and seed growth capacity [51,52]. The species *Trichoderma parareesei* also increased the productivity of canola plants under dry and salinity conditions [53]. Under drought conditions, Zhou et al. [54] also reported improved seedling growth of *Pinus tabulaeformis* after inoculation with fungi of the genus *Phoma*. Likewise, a positive effect was found under water stress following the inoculation of barley plants with the symbiont *Piriformospora indica*. Analysis of the plant proteome after *P. indica* inoculation showed an increase in proteins involved in energy metabolism, including photosynthesis, as well as in proteins involved in redox metabolism—mechanisms possibly related to barley’s tolerance to water stress [55]. Other benefits found in plants inoculated with fungi include superior plant resistance to temperature [56] and soil contaminated with heavy metals. For example, the endophytic fungus *Exophiala pisciphila* associated with the root of *Zea mays* generated a significant tolerance of the plant to Cadmium, leading to a decrease in phytotoxicity of the metal and an increase in maize growth [57].

### 2.2. Fungi as Biocontrol Agents of Plant Parasites

#### 2.2.1. Yeasts

Microorganisms colonize all parts of plants, from roots to shoots, including fruits. Some fungi remarkably establish a parasitic relationship with plants; however, it is also verified that the phytopathogens themselves can suffer the parasitism of another microorganism, which is called hyperparasitism [58]. In this way, the growth of phytopathogens is limited or inhibited. Therefore, hyperparasites can serve as biological control agents (BCA). They do this through (i) the secretion of secondary metabolites (which may have an antibiotic action), (ii) the production of VOCs and siderophores, (iii) competition for nutrients and space, or (iv) simply chemical communication [59]. Table 1 shows some examples of yeasts, their targets (inhibited plant parasites), and their action mechanisms.

The literature demonstrates that some yeasts manage to reduce the radial growth of pathogenic molds. *Metschnikowia pulcherrima*, for example, has already been identified as a potential parasite of the deteriorating fungus *Cladosporium cladosporioides* [15]. Similarly, the yeast *Pseudozyma aphidis* antagonizes phytopathogenic fungi of the genus *Phyllactinia* (powdery mildew), acting directly on their conidia via parasitism. Yeast causes conidial atrophy, collapse, and eventually the cleavage and death of powdery mildew. Dimorphic growth of the yeast *P. aphidis* was reported, which forms hyphae that grow around the conidia of *Phyllactinia* sp. and on its surface, using them as a source of nutrients [64]. Interestingly, another species of the genus *Pseudozyma* (in this case, *Pseudozyma churashimaensis*) showed antagonistic activity against the phytopathogenic bacteria *Xanthomonas axonopodi*, protecting pepper seedlings [65]. 

The fungus *Botrytis cinerea*, found on grapes (*Vitis vinifera*), is antagonized by the yeast *Metschnikowia pulcherrima*, which is also commonly found on fruits such as grapes and apples. This yeast produces pulcherrimine [11], a chemical complex formed by the linkage between iron and pulcherriminic acid [66]. In contrast to the usual siderophores, pulcherriminic acid, when bound to iron, precipitates as the pulcherrimine complex, which is not easily solubilized in an aqueous environment. Thus, there are strong indications that the yeast uses the chelator to obtain competitive advantages against fungi and also against bacteria through the depletion of iron as a nutrient [66,67,68]. Furthermore, strains of *M. pulcherrima* have already demonstrated chitinase activity [11]. This enzyme is capable of dissolving the fungal cell wall and is efficient in pest control [69].

The endophytic yeasts *Naganishia antarctica*, *Aureobasidium pullulans*, *Cryptococcus terrestris*, and *Filobasidium oeirense* also proved to be potential biocontrol agents against the pathogenic fungi *B. cinerea*, *Monilinia laxa*, *Penicillium expansum*, and *Geotrichum candidum* [70]. In the case of A. pullulans, the antifungal activity involved in the biological control of the pathogens was attributed to the alkaline protease produced by this yeast [71]. This enzyme breaks down proteins and may be applied to postharvest food. 

Similarly, strains of *Wickerhamomyces anomalus* isolated from grapes and other plants demonstrated strong antagonist activity against the molds *B. cinerea*, *Magnaporthe oryzae*, and *Roesleria subterranea*. In the secretome of this yeast, lytic enzymes with β-glucosidase, protease, amylase, and cellulase activity were identified [11]. *Wickerhamomyces anomalus* has also demonstrated its glucanase activity against the fungi *B. cinerea* and *P. expansum* on apples, *Penicillium digitatum* on oranges, and *Colletotrichum gloeosporioides* on papaya [72]. In another study, it was demonstrated that exoglucanase activity is necessary for the inhibition of the fungus *B. cinerea* by the yeast *W. anomalus* [73]. Thus, the antagonistic actions between yeasts and phytopathogenic fungi show clear signs of being related to enzymatic activities; in other words, cell wall lytic enzymes are valuable tools and can be associated with the effective fulfillment of this role of cell lysis [74].

#### 2.2.2. Molds

The control of parasites based mostly on chemical pesticides may cause problems like residues found in plants, fruits, and vegetables, besides serious environmental impacts [75]. For plant growth, especially in greenhouses, microorganisms have been studied as alternative sustainable control methods [76]. Fungi exhibit a vast array of characteristics and applications that offer numerous benefits to society, and their potential as biocontrol agents is of particular interest. In this case, they may be employed to manage plant diseases, presenting an attractive substitute for synthetic chemical products against various plant pathogens, such as oomycetes, nematodes, insects, and even other fungi [77]. Table 2 shows some examples of filamentous fungi, their targets (inhibited plant parasites), and their action mechanisms.

The control of *Duponchelia fovealis*, a European pepper moth that becomes a pest to some plants, was studied using fungal isolates from strawberry leaves [98]. A total of 517 fungal colonies were isolated from strawberry leaves, and 13 genera were identified. Eight endophytic fungi were evaluated in bioassays against *D. fovealis* and induced the highest mortality rates in its larvae. The root-knot nematode *Meloidogyne incognita* is a pest that interferes with the cultivation of sacha inchi (*Plukenetia volubilis* L.). Some fungi isolated from this species were explored as biocontrol agents against damage caused by *M. incognita* [99]. Compared to the control without any colonization, isolates colonized by *Trichoderma* and *Clonostachys* significantly reduced the number of galls induced by the nematode, besides allowing better root development in these plants. 

The potential antibacterial activity attributed to the genus *Trichoderma* is, in most cases, associated with its capability to produce secondary metabolites like pyrones, gliovirin, gliotoxin, and polyketides [100]. For instance, *Ralstonia solanacearum*, a severe bacterial pathogen of tomato plants, had its growth strongly inhibited by *Trichoderma pseudoharzianum* and *Trichoderma asperelloides* [101]. This effect was associated with secondary metabolites produced by both fungi, *Trichoderma*, which interfered with the cell morphology of *R. solanacearum* through rupture of cell walls, leakage of cell contents, and disintegration of the cell membrane. 

*Acremonium* sp. Ld-03 was isolated from the plant species *Lilium davidii* and demonstrated the capability of inhibiting the phytopathogenic fungi *Fusarium fujikuroi*, *B. cinerea*, and *F. oxysporum*. The inhibitory zones respectively observed were 33 mm (43%), 22 mm (56%), 30 mm (78%), and 30 mm (20%), with metabolites obtained from the ethyl acetate fraction [101]. In addition to its role as a biocontroller, Khan and colleagues [101] showed that *Acremonium* can act as a promoter of plant growth by increasing the production of the plant hormone IAA. Likewise, Chowdhary and Kaushik [96] evaluated *Acremonium* MPHSS-2.1 and MPM-2.1 against *B. cinerea*, *F. oxysporum*, and *Rhizoctonia solani*.

*Trichoderma*, a genus first described by Persoon in 1974, has been known for its biocontrol activities since 1932, as reported by Weindling [102]. This genus belongs to the *Hypocreaceae* family and can be found in various environments, particularly those with decomposing organic matter [103]. *Trichoderma* species can colonize plant roots and engage in a beneficial exchange of compounds, which offers protection against multiple phytopathogens [100]. For example, *T. virens* is known to produce trichodermamides, while *T. koningii* synthesizes koningins, which are compounds that possess antimicrobial and antifungal activity against plants [104,105]. The synthesis of hydrolytic enzymes and proteases, including exo- and endochitinases, chitinases, xylanases, glucanases, lipases, and endo- and exopeptidases, among others, has been extensively characterized in various species of *Trichoderma*. These enzymes exhibit antifungal properties, highlighting the potential of *Trichoderma* spp. as an effective biocontrol agent against fungal infections [101]. *Trichoderma atroviride* is a filamentous fungus commonly found in soil, especially in temperate regions. It exhibits optimal growth at a temperature of 25 °C and produces thin, hyaline colonies with inconspicuous aerial hyphae. After a period of 2 to 7 days, this fungus develops gray to dark green conidia [106] and shows the ability to control several phytopathogens, including other fungi [77].

The biocontrol activity of *Trichoderma harzianum* against the phytopathogenic fungi *Rhizoctonia solani* and *Pythium aphanidermatum* has been reported. Inocula with *T. harzianum* reduced the severity of rhizome rot and leaf blight in turmeric, diseases related to the so-called pathogens [107]. In addition to mycoparasitism, fungi of the genus *Trichoderma* also demonstrate antagonistic activities against nematodes and insects by different mechanisms: *T. harzianum* induced metabolic changes in tomato plants that attracted the parasitic wasp *Aphidius ervi*, which, in turn, neutralized the attack of the aphid *Macrosiphum euphorbiae* [108]. Also interestingly, the species *T. atroviride* not only induced systemic resistance in tomato plants against the nematode *Meloidogyne javanica*, but also the progeny of these plants inoculated with *T. atroviride* inherited this resistance [109].

The filamentous fungus *Clonostachys rosea* has been extensively researched as a biocontrol agent against many pathogenic fungi, such as *Alternaria dauci*, *Alternaria radicina*, *Botrytis cinerea*, *Botrytis aclada*, and *Helminthosporium solani* [85,110,111,112,113]. Furthermore, *C. rosea* has also been linked to the biocontrol of some nematodes and insects [114,115]. Regarding the mechanism of biocontrol by *C. Rosea*, studies of functional genetics and transcriptomic analyses with this fungus found some genes that seem to be important for mycoparasitism and biocontrol activity, such as those that encode proteins involved in the biosynthesis of secondary metabolites, enzymes that degrade pathogen cell walls, enzymes involved in complex carbohydrate metabolism, and membrane transporters related to drug resistance [116]. In addition, the gene encoding non-ribosomal peptide synthetase (NPS1) also appears to be important in biocontrol against *Fusarium graminearum*, a frequent pathogen in various crops such as maize, wheat, and beans [117].

## 3. Antagonistic Relationships

Despite the several examples above of mutualistic interactions, yeasts and molds may also be harmful to plants. In this regard, fungi can lead to critical losses in food production. However, from another point of view, this pathogenicity may be used to promote crops in some circumstances. This is the case with yeasts and molds that have been tested against weeds (as bioherbicides).

### 3.1. Pathogenicity

By threatening plant health, phytopathogenic fungi cause severe damage to world agricultural production [118,119]. Many of these fungi, such as those of the *Fusarium* and *Aspergillus* genera, can even produce toxins, posing a risk to human health when the infected plant is consumed [120]. Plants are under constant biotic stress [121], and the cell wall is an important barrier against invading pathogens. However, phytopathogenic fungi secrete a series of enzymes that degrade the plant cell wall by hydrolyzing its polysaccharides, thus allowing fungi to colonize the plant and obtain nutrients from its tissues [122,123,124].

Biotrophic pathogens, which do not lead to plant death and derive nutrients from living tissues, generally have a lower gene content for cell wall-degrading enzymes in their genome. In this way, damage to the host is more subtle, avoiding plant defense responses that can be triggered by the release of cell wall fragments [125,126]. On the other hand, necrotrophic and hemibiotrophic pathogens, with necrophilic stages of infection, have many enzymes that degrade the cell wall to guarantee their acquisition of nutrients [122,124].

Like many filamentous fungi, the yeast *Cryptococcus neoformans* also secretes cellulases, pectinases, laccases, and hemicellulases that degrade the cell wall of plant cells and provide their invasive behavior [127]. On the other hand, the colonizing behavior of some yeast species is often associated with their ability to form pseudohyphae, a growth morphology that allows them to explore new spaces through the elongation of the yeast cell. Thus, even in situations of nutrient scarcity, the pseudohypha enables the yeast to better scour for available nutrients [128,129]. It has also been reported that IAA produced by the plant can promote this invasive behavior in the form of pseudohyphae in *S. cerevisiae*. Actually, this phytohormone tends to prevent the proliferation of fungal cells, but in doing so, it ends up stimulating the morphological transition of the yeast [130]. Additionally, antagonistic yeasts may also be found in the rhizosphere. In some cases, they end up acting as a physical barrier, preventing the roots from obtaining nutrients from the soil [128]. 

Diseases caused by fungi, however, have long been addressed in the literature. Considering the focus of this work is mainly on the positive relationships for agriculture, the next section discusses how this pathogenicity can benefit crops against weeds. It is worth noting, though, that the use of fungi as bioherbicides has only been shown to be feasible because their antagonistic profile is species-specific.

### 3.2. Pathogenic Fungi as Bioherbicides

Weeds are a significant problem for food production, and their management is paramount for current agriculture. Thus, seeking herbicides that are less harmful to the environment and that provide effective control of weeds is highly desirable. And this can be achieved by bioprospecting microorganisms for the production of phytotoxic metabolites [131].

As stated before, endophytic fungi live within plant tissues without causing apparent damage to the host plant. However, they can produce bioactive compounds (which include volatile and non-volatile secondary metabolites) with phytotoxic properties, which can be exploited in agriculture for weed control [132,133]. Luckily, the bioactive compounds produced by fungi are quite specific against weeds, which minimizes the effects on cultivated plants [131,133].

Fungi can inhibit weeds’ growth by delaying nutrient absorption and preventing the synthesis of photosynthetic pigments and plant-growth hormones. Consequently, the production of reactive oxygen species and hormones mediated by plant stress increases. Some metabolites can cause disease, necrosis, and chlorosis, thereby inhibiting weed seed germination and growth [132,134] (Figure 2).

Phytopathogenic fungi are mainly among the genera *Alternaria*, *Botrytis*, *Colletotrichum*, *Fusarium*, *Helminthosporium*, and *Phoma* [135]. The fungus *Colletotrichum coccodes*, for example, was tested as a bioherbicide against the nightshade plant (*Solanum ptycanthum*), a troublesome weed in part because of its tolerance and resistance to certain conventional herbicides [136]. Several fungi of the genus *Phoma* show herbicidal activity against weeds, such as *Asparagus* sp., *Tricyrtis maculado*, *Beta vulgaris*, *Chenopodium album*, *Cirsium arvense*, and *Sonchus arvensis*, and this activity regards the several phytotoxic secondary metabolites produced by the fungus [137].

In the study by Portela et al. [131], the phytotoxicity of the fungus *Mycoleptodiscus indicus* UFSM-54 was tested on *Cucumis sativus*, *Conyza* sp., and *Sorghum bicolor*. The metabolites produced by the fungus impaired germination, initial growth, and the development of leaves in the three plant species. Interestingly, the metabolites obtained from *M. indicus* UFSM-54 were not toxic to rhizosphere-associated earthworms (*Eisenia andrei*). Souza et al. [134] selected fungi from the Brazilian Pampa biome capable of producing secondary metabolites with herbicidal activity. Of the fungi analyzed, 28 showed some level of phytotoxicity against the plant *C. sativus*. The most effective strain was sequenced and identified as belonging to the genus *Diaporthe*, previously reported as a bioherbicide producer [138,139].

Despite a few studies using yeasts as bioherbicides, there is some evidence of their potential to inhibit weed growth. Yeasts carrying enzymes such as lipases, cellulases, xylanases, pectinases, proteases, and peroxidases are capable of degrading the polysaccharides of weeds’ cell walls [140,141]. Species such as *Papiliotrema laurentii*, *Meyerozyma caribbica*, and *Rhodotorula mucilaginosa* isolated from insects and an Antarctic lagoon, for example, are capable of secreting these enzymes [140,142]. Moreover, metabolites such as VOCs and phenolic compounds, which can also be produced by plant-associated yeasts, interfere with the development mechanisms of weeds [132]. In this case, applying metabolites present in yeast cell extracts can be a strategy to control unwanted species.

## 4. Plant-Associated Fungi: Additional Biotechnological Perspectives

Since they are sessile living beings, plants have evolved by establishing a series of strategies to connect to the environment around them and survive adverse conditions. This includes a network of interactions with microorganisms that comprise the so-called plant microbiome [143]. These microorganisms, including fungi, can form complex associations with plants and play important roles in productivity, plant defense against pathogens, and overall plant health in natural environments. Because of this, researchers have explored the bioprospecting of fungi for the production of biofertilizers, bioherbicides, and biocontrol agents against phytopathogens. During this interaction with plants, several fungi have also been identified as producers of bioactive compounds and potential molecules for many applications of human interest and for distinct biotechnological fields [144] (Figure 3).

Plant-dwelling fungi have been linked to benefits for the plant, particularly in directly promoting plant growth and nutrient uptake [145]. Because of this, the isolation and production of bioinoculants based on plant microbiomes, such as fungi isolated from plants, are already being tested to increase the productivity of some crops. Moreover, fungi can also increase the nutritional value of forages, consequently benefiting livestock. The fungus *Sporormiella intermedia* increased the absorption of minerals when inoculated in the clover *Trifolium subterraneum*, while the fungus *Mucor hiemalis* increased the absorption of potassium and strontium in the grass *Poa pratensis*, demonstrating the potential of these fungi to improve the nutritional value of these forages [146].

The present review also addressed the role of fungi as biocontrol agents, eliminating or reducing the presence of pathogens. For this biocontrol action to occur, some mechanisms have been identified, such as competition between beneficial fungi and pathogens for nutrients and space, the secretion of volatile organic compounds by beneficial fungi, the production of lytic enzymes, mycoparasitism, and the induction of systemic resistance in the plant [110,147]. In agriculture, one of the applications under development in recent decades has been using fungi as mycopesticides, mainly due to the need for new alternatives that are less aggressive to the environment than chemical pesticides [147]. Currently, fungi belonging to the genus *Trichoderma* already make up bioformulations used to control fungal pathogens, mainly those that affect the root systems of crops. Among the species, the main ones employed are *T. atroviride*, *T. hamatum*, *T. harzianum,* and *T. viride* [110,148,149]. 

Last but not least, more than just being directly employed in crops (either as plant-growth-promoting or biocontrol agents), plant-associated fungi might show other biotechnological properties. For instance, yeasts of the genus *Pseudozyma*, which colonize many parts of plants, have been shown to be able to secrete a number of industrially useful products, such as lipases, glycolipids, and esterases [150]. Moreover, plant-associated microorganisms are widely recognized as great VOC producers, accounting for the attraction of pollinators to plants. In addition to being molecules with a wide range of industrial uses, VOCs may prevent or reduce plant diseases. For example, VOCs produced by *S. cerevisiae*, *W. anomalus*, *M. pulcherrima*, and *A. pullulans* are known to be efficient against the phytopathogens *B. cinerea*, *Colletotrichum acutatum*, *P. expansum*, *P. digitatum*, and *Penicillium italicum* [60,151]. The same holds true for some psychrotolerant yeasts that have also been reported to produce VOCs that inhibit fruit spoilage fungi. This is the case of *Leucosporidium scottii*, which produced VOCs that inhibited apple rot fungi [152], and the sake strain called Antarctic Candida, which is able to produce VOCs that inhibit the growth of the following phytopathogenic fungi: *P. expansum*, *B. cinerea*, *Alternaria alternata*, *Alternaria tenuissima*, and *Alternaria arborescens* [61]. Among the wide variety of VOCs produced by these fungi, phenylethanol is one of the most prominent and most commonly produced, being useful not only for agriculture and agroindustry but also as a flavoring agent and for household and personal care [3,46,60,61].

Fungi can also serve as biofactories. Some terpenoids from plant essential oils are recognized as bioherbicides. However, to be produced on a large scale and meet the demand for agricultural production, they would need large areas of cultivation and tons of biomass to generate the necessary amount [132,153,154]. In this case, through the metabolic routes of glycosides, yeasts can be used to produce these compounds from complex substrates containing polysaccharides [62,155,156]. Therefore, polysaccharide-rich waste hydrolysates (such as sugarcane bagasse, fruit peels, and algal biomass extract) can be used as feedstocks for the production of growth promoters, biocontrol agents, and bioherbicides with circular economy consistency [16,141,157].

## 5. Concluding Remarks and Future Perspectives

In nature, fungi and plants establish complex and dynamic ecological relationships. This interaction between the two biological kingdoms can benefit the plant, as the presence of some fungi can help with plant growth, pest control, increased nutrient absorption, or even plant resistance to different growth stresses. Through a better understanding of the biochemical bases of this mutualistic relationship, several molecules have already been discovered, such as phytohormones, vitamins, antibiotics, organic acids, siderophores, lytic enzymes, phenolic compounds, and volatile compounds, which play a key role in the benefits provided by fungi. In this sense, bioprospecting fungi that inhabit plants is a great starting point for developing fungal formulations that act as bioinoculants or in the biocontrol of pests, representing an emerging biochemical and biotechnological area. On the other hand, the fungus-plant interaction can be phytopathogenic, and many fungal diseases in plants are already well-known and studied. However, as the phytopathogenic profile of fungi is species-specific, this pathogenicity can benefit crops by using these fungi or their metabolites as bioherbicides against weeds—a broad biotechnological field still little explored. Finally, the metabolites resulting from the fungus-plant interaction represent a valuable source of biocompounds with applications in the health, cosmetics, pharmaceutical, and food industries.

Thus, there is a wide biotechnological area being discovered from the interaction between fungi and plants, and that can not only be applied to the improvement and defense of plant cultures. Many compounds arising from this interaction may also represent new molecules of interest to the food, cosmetics, and drug industries [16,144]. Indeed, the large-scale biotechnological application of these new fungi and/or their metabolites requires (i) a biochemical understanding of the fungus-plant interaction (the fungus-pathogen interaction and the compounds generated with and during this interaction), (ii) tests with the application of fungal inoculum in plant cultures in the laboratory and in the field, also evaluating the environmental ecotoxicity, and (iii) the analysis of the bioactive potential of the metabolites generated from this fungus-plant interaction.

## Figures and Tables

**Figure 1 plants-12-02688-f001:**
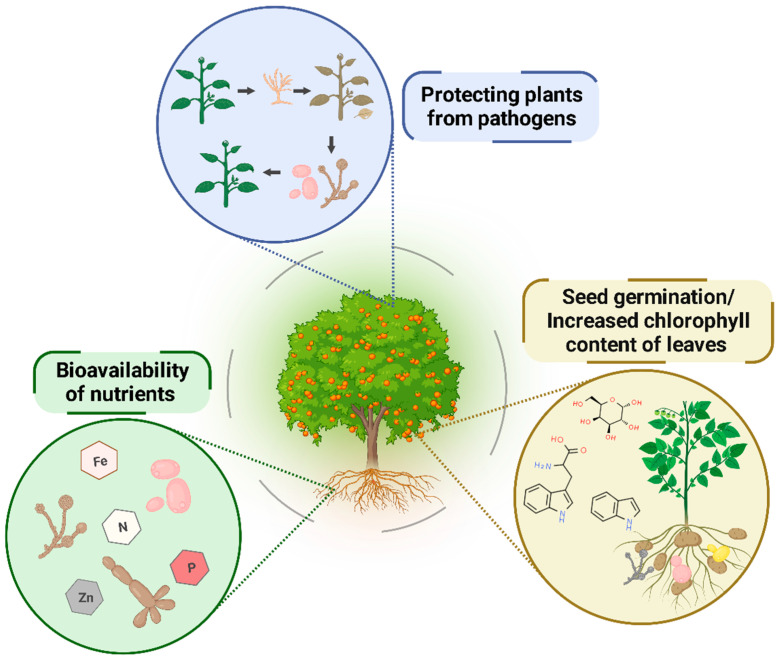
Mutualistic interaction between fungi and plants. Yeasts and molds can act (i) by making macro- (nitrogen and phosphorus) and micronutrients available to the plants; (ii) by producing phytohormones, which stimulate faster seed germination and higher chlorophyll content on the leaves; and (iii) by protecting plants against pathogens.

**Figure 2 plants-12-02688-f002:**
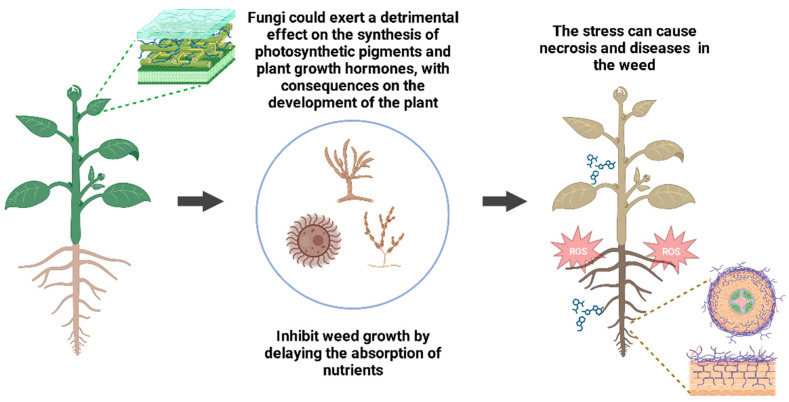
Action mechanisms of pathogenic fungi on weed control. Molds and yeasts can inhibit weed growth by (i) preventing the synthesis of photosynthetic pigments, (ii) delaying nutrient uptake, and (iii) inducing necrosis of plant tissues by reactive oxygen species and stress hormones.

**Figure 3 plants-12-02688-f003:**
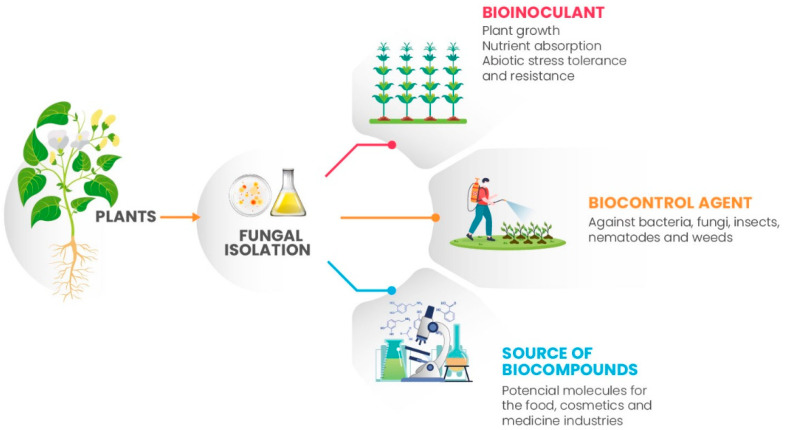
Bioprospecting plant-associated fungi. Fungi isolated from plants have shown the potential to be employed (i) as bioinoculants (which help directly with crop growth and nutrient uptake or indirectly through increased tolerance to abiotic stresses, mainly salinity, drought, and soil resistance with heavy metals), (ii) as biocontrol agents (which have an antagonistic activity to bacteria, fungi, insects, nematodes, and weeds), or (iii) as a source of biocompounds (which produce potential molecules for many applications of human interest).

**Table 1 plants-12-02688-t001:** Yeasts, their targets (inhibited parasites), and their action mechanisms.

Yeast	Target Organism	In Vivo Application	Compound	Mechanism	Reference
*Metschnikowia pulcherrima*	*Botrytis cinerea*	Yeast cell suspension on apple	Pulcherrimin complex and Chitinase	Nutrient competition (Iron) and cell wall degrading enzyme	[11]
*Aureobasidium pullulans*	*Botrytis cinerea, Colletotrichum acutatum*, *Penicillium italicum*, and *Penicillium. digitatum*	Yeast cell suspension on orange	Phenethyl alcohol, 1-Butanol, 1-Propanol	Fungal growth inhibition	[60]
*Candida sake*	*Penicillium expansum*, *Botrytis cinerea*, *Alternaria alternata* CBS916.96, *Alternaria tenuissima* CBS 124.277, and *Alternaria arborescens* CBS 102.605	Yeast cell suspension on apple	Phenylethyl alcohol, 2-Phenylethyl acetate, 3-Methylbutyl hexanoate, 3-Methylbutyl pentanoate, 2-Methylpropyl hexanoate	Fungal growth inhibition	[61]
*Metschnikowia andauensis*	*Spodoptera littoralis*	Larval feeding on living yeasts	3-Methylbutan-1-ol, α-Terpineol, 3-Methyl butanoic acid, Isobutanoic acid	Larval feeding avoidance	[62]
*Saccharomyces cerevisiae*	*Spodoptera littoralis*	Larval feeding on living yeasts	Methyl palmitate, Heptan-1-ol, Ethyl octanoate, Ethyl decanoate, Ethyl hexanoate	Larval feeding avoidance	[62]
*Meyerozyma guilliermondii* KL3	*Penicillium digitatum* DSM2750 and *Penicillium* *expansum* DSM62841	Yeast cell suspension	Phenylethyl alcohol, 2-ethylhexanol, D-limonene, benzaldehyde, 2-methylbutanol, and 3-methylbutyl hexanoate	Fungal growth inhibition	[63]

**Table 2 plants-12-02688-t002:** Filamentous fungi, their targets (inhibited parasites), and their action mechanisms.

Fungi	Target Organism	Action Mechanism	Reference
*Trichoderma atroviride*	*Fusarium avenaceum*, *Fusarium culmorum*	Expression of defense genes: PR2, GST1, PAL, and STS	[78]
	*Fusarium avenaceum*, *Fusarium culmorum*	No elucidated	[79]
	*Phytophthora cinnamomic*	Competition by nutrients	[80]
	*F. avenaceum*	endo-b-1,3-glucanases; 6-n-pentyl-6H-pyran-2-one	[79]
	*Rhizoctonia solani*, *Fusarium oxysporum*	6-pentyl-α-pyrone (6-PP)	[81]
	*B. cinerea*, *Sclerotium cepivorum*, and *Colletotrichum lindemutianum*	Expression of defense genes: Tal6	[82]
	*F. graminearum*	Expression of defense genes: Vel1	[83]
*Phlebiopsis gigantea*	*Heterobasidion* spp.	Competition for growth and establishment	[84]
*Clonostachys rosea*	*Sclerotinia sclerotiorum, Bemisia tabaci*	Secretion of metabolites and enzymes	[85]
*Trichoderma harzianum* HK-61	Cucumber mosaic virus	Defense response: trichokonins	[86]
*Ampelomyces quisqualis*	*Podosphaera fusca*	Secretion of enzymes, metabolites, and competition by nutrients	[87]
*Trichoderma harzianum* GIM 3.442	*Mucor circinelloides*, *Aspergillus flavus*, *Aspergillus fumigatus*	Protease P6281	[88]
*Trichoderma harzianum* CCTCC-RW0024	*Fusarium gramineraum*	Chitisnase, β (1,3) glucanase, H-Benzopyrano [3,4-b]pyridin-5-one,-amino-1,2,3,4-tetrahydro	[89]
*Sporothrix flocculosa*	*Sphaerotheca pannosa* and *Sphaerotheca fuliginea*	Production of (Z)-9-heptadecenoic, which causes Disruption in the fluidity of the cellular membrane, resulting in the release of molecules such as electrolytes and proteins	[90]
*Kloeckera apiculate*	*Penicillium italicum*	Competition for nutrients	[91]
*Trichoderma harzianum*	*Fusarium oxysporum*	β-1,3-glucanolytic, and Chitinases	[92]
*Trichoderma harzianum*	*Pythium ultimum*	Chlorophyll, flavonoids, and antioxidant	[93]
*Pseudozyma aphidis **	*Botrytis cinerea*	Enzymes such as chitinase, protease lipase, and cellulase	[94]
*Trichoderma atroviride*	*Alternaria solani* and *Rhizoctonia. solani*	Swollenin	[95]
*Acremonium* sp. MPHSS-2.1	*Sclerotinia sclerotiorum*	1-heptacosanol and 1-nonadecane	[96]
*Pseudozyma flocculosa **	*Blumeria graminis*	Expression of genes: pf02826, pf00303 and pf02382. Dissemination and sequestration of nutrient	[97]

* *Pseudozyma* is a genus of dimorphic fungi (they present hyphal and yeast-like growth).

## Data Availability

Not applicable.

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
