# Peer review of "Biochemical and Biotechnological Insights into Fungus-Plant Interactions for Enhanced Sustainable Agricultural and Industrial Processes"

_plants, 2023, doi:10.3390/plants12142688_

Round 1
Reviewer 1 Report
Dear Author,
Please see the comments below:
1. Author advised to add one comprehensive table for the mold and yeast, their various application, major compounds, mechanism etc.
2. Table 1 mentioned the "Filamentous fungi, their inhibited parasites, and their action mechanisms" however it only describing the Trichoderma spp. mention other fungi also.
3. Title of the manuscript is very broad and mention about the insight but its providing very superficial information.
4. What is the criteria used to describe these fungi and yeast. Author can use search engine to accumulate data to no. of publication, fungi for the a given period and insight with clear objective.
5. Description about Volatile and non-volatile compound is needed.
6. add major bioactive compound and their description in biotechnological applications.
7. Add separate conclusion section.
Author Response
We are thankful for the reviewer's comments (below, in blue italics). They were extremely helpful in improving our manuscript. Our responses are given right after each comment.
- Author advised to add one comprehensive table for the mold and yeast, their various application, major compounds, mechanism etc.
Response: We agree with the reviewer's comment. We have added a table for yeasts.
- Table 1 mentioned the "Filamentous fungi, their inhibited parasites, and their action mechanisms" however it only describing the Trichoderma spp. mention other fungi also.
Response: The reviewer is right; we have improved Table 1.
- Title of the manuscript is very broad and mention about the insight but its providing very superficial information.
Response: We agree with the reviewer's comment. We have made the title more representative.
- What is the criteria used to describe these fungi and yeast. Author can use search engine to accumulate data to no. of publication, fungi for the a given period and insight with clear objective.
Response: We thank the reviewer for this comment. Fungus-plant interactions are quite a wide subject. When we were conceptualizing the manuscript, we did not want to restrict the period for the reference to be consulted. Our intention was to address the most representative studies in plant-growth-promoting strategies, bioherbicides, and plant-associated fungi as microbial factories. Anyway, in the last paragraph of the introduction, we have now defined "molds/filamentous fungi" and "yeast" (highlighted in yellow in the revised manuscript).
- Description about Volatile and non-volatile compound is needed.
Response: We have now defined "volatile" and "non-volatile" compounds in the last paragraph of the introduction (highlighted in yellow in the revised manuscript).
- add major bioactive compound and their description in biotechnological applications.
Response: We thank the reviewer for this suggestion. We have added this information in lines 528–531 of the revised manuscript (highlighted in yellow).
- Add separate conclusion section.
Response: The reviewer is right; we added a separate conclusion section.
Reviewer 2 Report
In this review the authors discuss the fungus-plant interaction based on reported results. It is well written and interesting. As a review provide an analysis of the current works. All the reference are appropriate. In my opinion, the manuscript is suitable for publication.
Author Response
In this review the authors discuss the fungus-plant interaction based on reported results. It is well written and interesting. As a review provide an analysis of the current works. All the reference are appropriate. In my opinion, the manuscript is suitable for publication.
Response: We are thankful for the kind words of the reviewer.
Reviewer 3 Report
The manuscript plants-2491457 entitled « Biochemical and Biotechnological Insights into the fungus-plant Interactions” discussed the mutualistic and antagonistic relationships in fungus-plant interaction. This subject is well documented in the literature. This review has some issues. Please see the specific comments below.
-The English language must be revised thoroughly.
Abstract:
Line 22-23: “Moreover, even when the plant pathogens are the fungi themselves, there is still something that humanity might take advantage of”. This phrase is out of context since you are talking about plant-fungi interactions.
-The objective should be re-formulated to highlight the importance of the work and what it offers plus in the field.
-The keywords are representative key elements of what it’s written in the abstract. However, I find it useless to put the words; chelator; VOCs; and yeast that are not presented in the abstract.
Introduction
-At the beginning of the introduction part you should highlight the type of fungi since in line 35 you referred them as microorganisms.
-This part must be extended and more information about the study should be added. Moreover, the working hypothesis must be revised.
Context:
I propose to add an introductive paragraph before starting with mutualistic and antagonistic relationships.
-In Table 1 you provided examples linked to the Trichoderma genus. However, the literature is rich with other reported genera.
-This review contains one short table and one figure. Additional tables and figures are necessary to enrich your work.
-Figure 1 is misplaced after the conclusion part. The caption should be revised thoroughly.
-Line 71: Replace “2.1. Plant growth promoted by yeasts and molds” with “Plant Growth Promotion”.
Conclusion:
This part must be revised thoroughly (it is more pertinent to move this part in the text). You should Summarize the main points discussed in your manuscript. Moreover, future directions should be added.
The English language must be revised thoroughly.
Author Response
We appreciated the reviewer's comments (below, in blue italics). Undoubtedly, they helped improve our manuscript. Our responses are given right after each comment.
-The English language must be revised thoroughly.
Response: I asked an English native-speaker friend to proofread it, and I incorporated his few corrections to the text.
Abstract:
Line 22-23: "Moreover, even when the plant pathogens are the fungi themselves, there is still something that humanity might take advantage of". This phrase is out of context since you are talking about plant-fungi interactions.
Response: We thank the reviewer for this comment. However, we believe that humanity does benefit (take advantage of) from "fungi-plant interactions". In fact, this is one of the main points of our manuscript. Anyhow, we have clarified this in the abstract (please see the changes highlighted in yellow in lines 23–24 of the revised manuscript).
-The objective should be re-formulated to highlight the importance of the work and what it offers plus in the field.
Response: The reviewer is right. We reformulated the objective.
-The keywords are representative key elements of what it's written in the abstract. However, I find it useless to put the words; chelator; VOCs; and yeast that are not presented in the abstract.
Response: We thank the reviewer for this comment. We kept the word "yeast" because now it appears in the abstract, but we suppressed "chelator" and "VOCs".
Introduction
-At the beginning of the introduction part you should highlight the type of fungi since in line 35 you referred them as microorganisms.
Response: We appreciated the reviewer's comment. We believe this has been clarified (please see the changes highlighted in yellow in lines 35–37 of the revised manuscript).
-This part must be extended and more information about the study should be added. Moreover, the working hypothesis must be revised.
Response: We have improved the manuscript's introduction. We believe it now meets the reviewer's comments (please see the changes highlighted in yellow in the revised manuscript).
Context:
I propose to add an introductive paragraph before starting with mutualistic and antagonistic relationships.
Response: These introductive paragraphs have been added.
-In Table 1 you provided examples linked to the Trichoderma genus. However, the literature is rich with other reported genera.
Response: We agree with the reviewer. This table has been improved.
-This review contains one short table and one figure. Additional tables and figures are necessary to enrich your work.
Response: The reviewer is correct. We added one more table, improved the existing one, and created two additional figures.
-Figure 1 is misplaced after the conclusion part. The caption should be revised thoroughly.
Response: The former conclusion has now been incorporated into the main text as one of its subtitles (now entitled "Plant-associated fungi: additional biotechnological perspectives"). Thus, we believe the figure has now fitted better in the manuscript. The figure caption has been revised.
-Line 71: Replace "2.1. Plant growth promoted by yeasts and molds" with "Plant Growth Promotion".
Response: The replacement has been made.
Conclusion:
-This part must be revised thoroughly (it is more pertinent to move this part in the text).
Response: This part has been incorporated into the main text as one of its subtitles (now entitled "Plant-associated fungi: an additional biotechnological perspective"). Moreover, a new conclusion section has been created in order to summarize the main points discussed in the manuscript and to give some future perspectives.
-You should Summarize the main points discussed in your manuscript. Moreover, future directions should be added.
Response: We are thankful for all the reviewer's comments. A new conclusion section has been created in order to summarize the main points discussed in the manuscript and to highlight some future perspectives.
Round 2
Reviewer 1 Report
NA
Reviewer 3 Report
The authors ameliorated the manuscript quality according to the reviewer's suggestions. I think that the manuscript will be suitable for publication in Plants after a minor revision of the English language.
A moderate revision of the English language is required.